# Multiplex short tandem repeat profiling of immortalized hepatic stellate cell line Col-GFP HSC

**Steffen K. Meurer**[1], **David A. Brenner**[2], **Ralf Weiskirchen**[1]*

1 Institute of Molecular Pathobiochemistry, Experimental Gene Therapy and Clinical Chemistry (IFMPEGKC), RWTH University Hospital Aachen, Aachen, Germany, 2 Department of Medicine, University of California San Diego, La Jolla, CA, United States of America

* rweiskirchen@ukaachen.de

**Data Availability Statement:** The determined STR profile of Col-GFP-HSC was deposited in the Cellosaurus database under accession no. CVCL_B7MI.

## Abstract

Misidentification, cross-contamination and genetic drift of continuous animal cell lines are persistent problems in biomedical research, leading to erroneous results and inconsistent or invalidated studies. The establishment of immortalized hepatic stellate cell line Col-GFP HSC was reported in *PLoS One* in the year 2013. In the present study a multi loci short tandem repeat signature for this cell line was established that allows for unique cell line authentication.

## Introduction

In the latest version of the register of misidentified cell lines published on 8 June 2021 by the International Cell Line Authentication Committee (ICLAC) currently lists 576 misidentified cell lines [1]. In most cases, this problem results from poor and inattentive handling of cell lines combined with a lack of routine quality control. To mitigate wasted research efforts and false claims in the literature, many biomedical research funding entities, such as the NIH have announced review criteria to enhance reproducibility of research findings through increased scientific rigor and transparency. In particular, special attention is given to authentication of biological materials such as cell lines [2]. Similarly, several scientific journals have included instructions or requirements for cell line authentication in their author guidelines. *PLoS One* for example notes in respective guidelines "*Cell line authentication is recommended–e.g., by karyotyping, isozyme analysis, or short tandem repeats (STR) analysis–and may be required during peer review or after publication*" [3].

The current recommended method for cell authentication is short tandem repeat (STR) profiling [4]. A STR, also known as a microsatellite, is a non-coding, short DNA sequence composed of a number of variable repeats of two to ten nucleotides in length. In regard to human cell lines, STR profiling has been identified in 2010 as an important step to eradicate incorrectly identified cell lines [5]. Some years later, efforts were also started to develop strategies for authentication of mouse cell lines. In a landmark paper published by the National Institute of Standards and Technology (NIST), a multiplex PCR assay based on nine STR markers was established that is suitable to authenticate individual mouse cell lines [6]. Several

**Funding:** The laboratory of R.W. is supported by the German Research Foundation (grants WE2554/13-1, WE2554/15-1, and WE2554/17-1). The funder had no role in study design, data collection and analysis, decision to publish, or preparation of the manuscript.

**Competing interests:** The authors have declared that no competing interests exist.

years later, a Consortium for Mouse Cell Line Authentication consisting of 12 participating laboratories was formed representing institutions from academia, industry, biological resource centers, and government with the aim to validate an extended STR marker set [7]. In the mentioned study, the respective multiplex PCR panel of mouse STR markers consisting of 19 loci was shown to be capable of discriminating at the intra-species level between 50 commonly used mouse cell lines [7].

Nowadays, reputable cell repositories including the American Type Culture Collection (ATCC), German Collection of Microorganisms and Cell Cultures (DSMZ), RIKEN Cell Bank (RCB), and the Japanese Collection of Research Bioresources (JCRB) have established authentication guidelines and maintain cell line databases that include thousands of individual cell lines. When depositing a new cell line in one of these repositories, the cell line undergoes rigorous identity and quality controls, notably authentication by STR profiling before making them available to the scientific community. As such, cell banks should be able to provide a certified DNA STR profile for each cell line distributed.

Several years ago, we generated and characterized a novel continuous cell line termed Col-GFP HSC. This cell line was derived from primary hepatic stellate cells (HSC) that were isolated from a transgenic mouse expressing green fluorescent protein (GFP) under control of the collagen α1(I) promoter/enhancer [8]. The cells were immortalized by infection with a lentivirus vector containing the Simian virus large T antigen (SV40T) and a hygromycin resistance cassette [8]. Because these cells are responsive to pro-fibrogenic stimuli, such as PDGF or TGF-β1, and are able to activate intracellular signalling pathways including Smads and MAP kinases, this cell line is a promising tool which can be used to investigate special issues of fibrogenic signaling. We have received many requests to offer this cell line to other laboratories, however, a reference genetic profile for this new cell line was not yet established.

This study reports an STR profile for this cell line including the electropherogram images that can be used as a unique fingerprint of Col-GFP HSC. The STR profile consists of the 19 species-specific markers that were proposed by the Consortium for Mouse Cell Line Authentication for authentication of mouse cell lines. This newly established reference profile can now be included in subsequent publications or grant applications to fulfill the need for authentication of key biological resources when working with Col-GFP HSC.

## Materials and methods

### Cell culture

Col-GFP HSC cells were routinely propagated in 10 cm Petri dishes and cultured in Dulbecco's modified Eagle's medium (DMEM) supplemented with 10% fetal bovine serum (FBS), 1 x Penicillin/Streptomycin, 1 mM sodium pyruvate, and 2 mM L-Glutamine. Medium was exchanged every second day and detachment of cells for subculturing was completed using Accutase solution (#A6964, Sigma-Aldrich, Merck, Darmstadt, Germany). The cells were isolated at 80% density for STR profiling, which is about $8 \times 10^6$ cells per 10 cm dish.

### Short tandem repeat profiling

STR profiling and interspecies contamination for Col-GFP HSC was performed by a commercial service provided (IDEXX, Kornwestheim, Germany). In the respective assay, (i.e., the Cell-Check Mouse 19 panel), the cells were genotyped using a panel of 19 mouse STR markers that have been described previously [6, 7]. The respective primers that are used in this assay for individual markers are given in Table 1.

This system establishes a genetic microsatellite marker profile and allows the detection of as little as 5% interspecies cross-contaminations. Individual markers were analyzed by

**Table 1. Primers used in the mouse multiplex PCR assay.**

| Marker | Forward Primer (5'→3') | Reverse Primer (5'→3') |
|---|---|---|
| 1–1 | CCCTTCACTCCTTCATTCCA | TGAGCCTAAGGACCTGGACA |
| 1–2 | TCTTTAAAAATCAAACAGGCAAA | GGGGAGGTTGGGGTGTATAA |
| 2–1 | ACAACTTCAAACTTTGCTGGAA | TCCTGCTTAGCTTGTCATTTCC |
| 3–2 | TGAGCTACCATGTGGGTACA | CACACACACACACAAAGATGGA |
| 4–2 | AAGCTTCTCTGGCCATTTGA | TTCATAAACTTCAAGCAATGACA |
| 5–5 | CGTTTTACCTGGCTGACACA | TGGTTTAAAACTCAATACCAAACAA |
| 6–4 | TTTGCAACAGCTCAGTTTCC | AATCGCTGGCAGATCTTAGG |
| 6–7 | AGTCCACCCAGTGCATTCTC | CATGTGGCTGGTATGCTGTT |
| 7–1 | ACACCAGTTGGGGACTAGGA | AACTGGTGCTGGGTCTGAAC |
| 8–1 | AGTAATATCCTGGTCCTGGCC | GAGCTCACTATGTAGCTATTGGA |
| 9–2* | GGATTGCCAAGAATTTGAGG | TCCTGAGTTGTGGACAGGGTTA |
| 11–2 | AAGGCAGGGGAATTCACAGT | TCTCACCATTGCAGTCCTGA |
| 12–1 | TTTCAAAATTGTCATTGAACACA | TGGTCCTTCAGTATCATCCTTG |
| 13–1 | AACTCCCACTGCTTCTTGGT | GCCAAAGTAAACCCTTCTCCC |
| 15–3 | TCTGGGCGTGTCTGTCATAA | AGGCTATCTAAACTTGACATGCT |
| 17–2 | CTCTTCTCCATCCGTCCATC | ATGGCTCTTGCCACAAATCT |
| 18–3 | TCTTTCTCCTTTTGTGTCATGC | GCTAAATAACTAAGCAAGTGAACAGA |
| 19–2 | AGGCTAGCACTGTTCCTTGT | ACTCAGCACCTTCCATCCTG |
| X-1 | AACAAAAATGTCCCTCAATGC | AAGGTATATATCAAGATGGCATTATCA |

* Information of primer pair for marker 9–2 was taken from [6], all other primer pairs were taken from [7]. Forward primers are given without fluorescent dyes at their 5' ends and reverse primers are given without the "PIGtail" sequences (a 7-nucleotide tag with sequence GTTTCTT) that are used in this assay to promote complete adenylation. For more details of this multiplex PCR assay please refer to the original publication in which this assay was established [7].

GeneMapper software 6. The determined STR profile of Col-GFP-HSC was deposited in the Cellosaurus database under accession no. CVCL_B7MI. An STR similarity search was performed using the CLASTR 1.4.4 matching tool, which can be found on the Cellosaurus STR database (release 41.0) [9]. The settings for the search were set to the following: Scoring algorithm: Tanabe, Mode: Non-empty markers, Score filter: 70%, and Min. Markers: 8. This scoring algorithm gauges the similarity of two samples. It is simply defined as: Percent match = (number shared alleles x 2) / (total number of alleles in the questioned profile + total number of alleles in the reference profile) [10].

## Results and discussion

Cross-contamination and misidentification of mammalian cell cultures is widespread, leading to thousands of misleading and potentially erroneous published papers [5]. In 2017 a conservative estimate found that 32,755 articles reported research results with misidentified cells [11]. Moreover, the mentioned study revealed that over 92% of these 'contaminated' papers is cited at least once, spreading potential misleading information as a 'secondary contamination' to the scientific community. Therefore, establishing a cell line's identity prior to performing experiments is essential to conduct valid and reproducible research.

The Col-GFP HSC cell line was established nearly ten years ago [8]. The cell line was derived from primary hepatic stellate cells (HSC) isolated from a Col-GFP transgenic reporter mouse model. Immortalization was achieved by infecting respective cells with a lentiviral vector containing the SV40T and a hygromycin resistance cassette. As such, the cells express green fluorescent protein, SV40T, and a characteristic set of HSC marker genes including α-

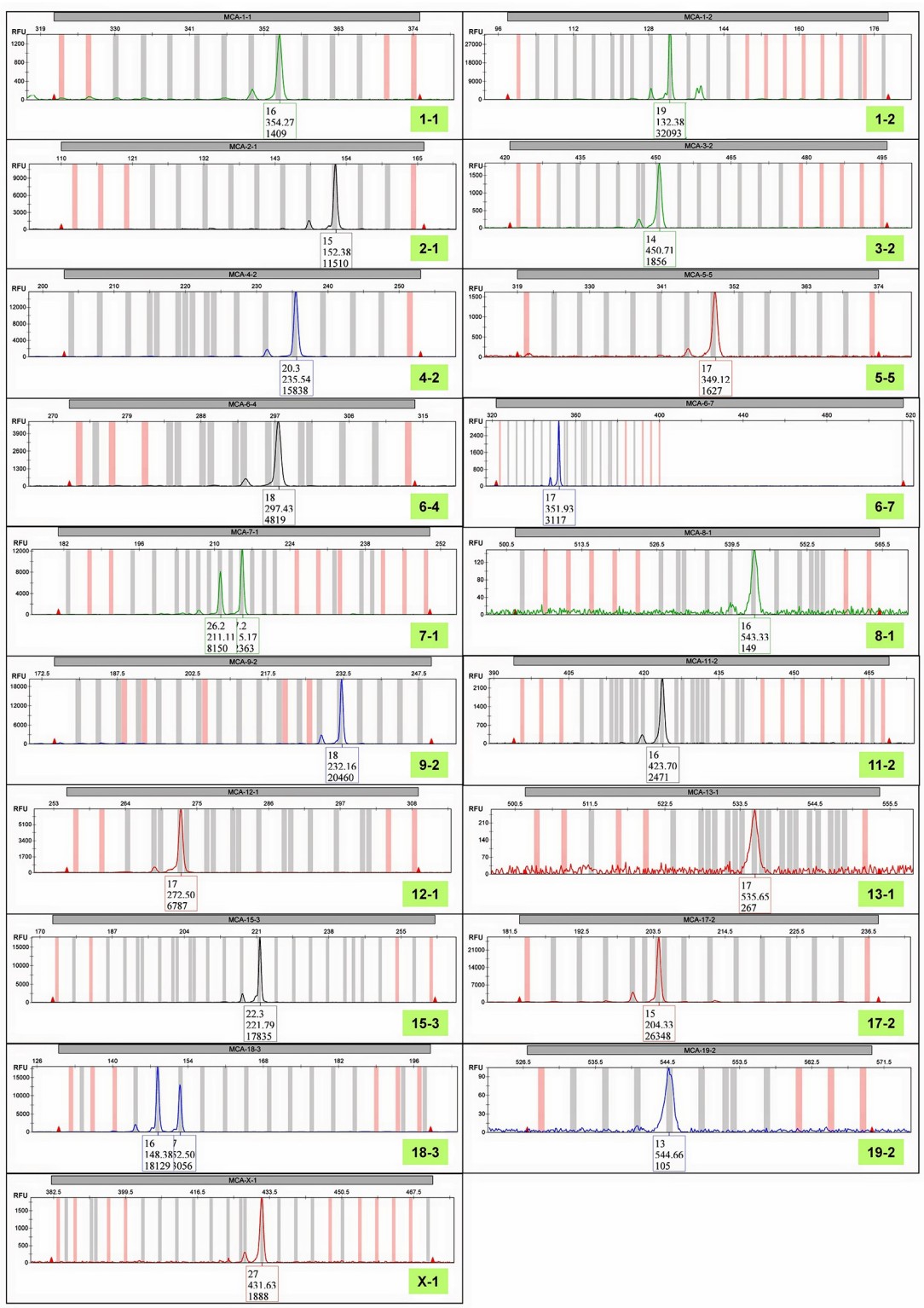

**Fig 1. Electropherograms for Col-GFP HSC resulting from analysis of 19 consensus markers.** DNA from Col-GFP HSC cells was isolated and the genetic profile determined using the STR markers 1–1, 1–2, 2–1, 3–2, 4–2, 5–5, 6–4, 6–7, 7–1, 8–1, 9–2, 11–2, 12–1, 13–1, 15–3, 17–2, 18–3, 19–2, and X-1, respectively. In this assay, the threshold of the assay was set to 100 relative fluorescence units (RFU).

**Table 2. Short tandem repeat (STR) marker profile of Col-GFP HSC.**

| SN | STR Marker | Known Allele Range* | Chromosomal Location | Allele |
|----|-----------|---------------------|----------------------|--------|
| 1 | 1–1 | 10–19 | 1 | 16 |
| 2 | 1–2 | 12–30.1 | 1 | 19 |
| 3 | 2–1 | 8–17.1 | 2 | 15 |
| 4 | 3–2 | 9–25 | 3 | 14 |
| 5 | 4–2 | 13–23.3 | 4 | 20.3 |
| 6 | 5–5 | 11–22 | 5 | 17 |
| 7 | 6–4 | 12.2–21 | 6 | 18 |
| 8 | 6–7 | 11–26 | 6 | 17 |
| 9 | 7–1 | 19.2–33.2 | 7 | 26.2,27.2 |
| 10 | 8–1 | 6–19 | 8 | 16 |
| 11 | 9–2 | 7.1–19 | 9 | 18 |
| 12 | 11–2 | 12–26.2 | 11 | 16 |
| 13 | 12–1 | 15–24.1 | 12 | 17 |
| 14 | 13–1 | 11–20.1 | 13 | 17 |
| 15 | 15–3 | 12–31.3 | 15 | 22.3 |
| 16 | 17–2 | 11–22 | 17 | 15 |
| 17 | 18–3 | 13–28 | 18 | 16,17 |
| 18 | 19–2 | 10–16 | 19 | 13 |
| 19 | X-1 | 15.2–37 | X | 27 |

* The term "known allele range" corresponds to the number of repeats that might exist at the analyzed polymorphic marker sites. The allele range for STR marker 9–2 was taken from [6], all other allele ranges were taken from [7].

smooth muscle actin (α-SMA), vimentin, desmin, and collagen type I [8]. Moreover, the cells are sensitive towards ligands involved in fibrosis and can be indirectly used to monitor the regulation of Collagen IαI expression [12]. Therefore, this cell system is an ideal experimental tool for cell-tracking experiments, co-culture systems or any other kind of studies in which cells of HSC origin should be investigated. However, the mentioned features are not necessarily specific for this cell line. The viral oncogene SV40T is a widely used agent to obtain immortalization or conditional reprogramming in primary cells [13]. Similarly, GFP and its derivatives have gained widespread use as a reliable and easily traceable reporter of gene expression or cellular structures in individual eukaryotic cells [14, 15].

Therefore, and to fulfill actual requirements and standards for authentication of a cell line, we established a genetic profile for Col-GFP HSC that is based on 19 species-specific STR markers (Fig 1, Table 2).

To determine if the genetic profile for Col-GFP HSC was unique, the CLASTR search tool in the Cellosaurus database was utilized. The profile search using the Cellosaurus STR database (release 41.0) containing 77 mouse cell lines revealed that Col-GFP HSC has the highest homology to cell lines AT-3 (CVCL_VR89; 82.93%), YUMMER1.7-H2B-GFP5 (CVL_A2AY, 80.00%), DC2.4 (CVCL_I409, 78.05%), MS1 (CVCL_6502, 78.05%), YUMM1.7 (CVCL_JK16, 76.92%), MCA-205 (CVCL_VR90, 76.19%), and MLTC-1 (CVCL_3544, 73.17%), respectively (Table 3).

The determined STR profile provides the power to discriminate Col-GFP HSC from other cell lines of mouse origin and is also suitable to detect interspecies contamination [6, 7, 16]. In the future, this STR profile should be used as a reference for authentication of this line. As such, it is another small step to implement the efforts to enhance reproducibility of research

**Table 3. CLASTR matching algorithm results for mouse STR profiles most similar to Col-GFP HSC to known STR profiles listed in Cellosaurus database\*.**

| STR Marker | Col-GFP HSC | AT-3 | YUMMER1.7-H2B-GFP5 | DC2.4 | MS1 | YUMM1.7 | MCA-205 | MLTC-1 |
|---|---|---|---|---|---|---|---|---|
| 1–1 | 16 | 16 | 16,17 | 17 | 17 | 16,17 | 16 | 16 |
| 1–2 | 19 | 19 | 19 | 19 | 19 | 19 | 19 | 19,20 |
| 2–1 | 15 | 16 | 16,17 | 16 | 16 | 16 | 16 | 16 |
| 3–2 | 14 | 14 | 14 | 14 | 14 | 14 | 13,14 | 14,15 |
| 4–2 | 20.3 | 20.3 | 20.3 | 21.3 | 20.3 | 20.3 | 20.3 | 20.3,21.3 |
| 5–5 | 17 | 17 | 17 | 17 | 17 | 17 | 17 | 17 |
| 6–4 | 18 | 18,19 | 18 | 18 | 18 | 18 | 18 | 18 |
| 6–7 | 17 | 15 | 17 | 17 | 16,17 | 17 | 14,15 | 16 |
| 7–1 | 26.2,27.2 | 26.2,27.2 | 27.2 | 26.2,27.2 | 26.2 | 27 | 26.2 | 26.2 |
| 8–1 | 16 | 16 | 16 | 16 | 16 | 16 | 16 | 16 |
| 11–2 | 16 | 16 | 16 | 16 | 16 | 16 | 16 | 16 |
| 12–1 | 17 | 17,18 | 17 | 17 | 17 | 17 | 17,18 | 17 |
| 13–1 | 17 | 17 | 17 | 17 | 17 | 17 | 17 | 17 |
| 15–3 | 22.3 | 22.3 | 22.3 | 22.3,23.3 | 22.3 | 22.3 | 22.3 | 22.3 |
| 17–2 | 15 | 15 | 16 | 16 | 15,16,17 | 16 | 15 | 15 |
| 18–3 | 16,17 | 16 | 16 | 16 | 16 | 16 | 16 | 16 |
| 19–2 | 13 | 13 | 13 | 13 | 13 | 13 | 13,14 | 13 |
| X-1 | 27 | 27 | 27 | 27 | 27 | 27 | 27 | 28 |

\* The search was done on April 15, 2022 at the Cellosaurus STR database (release 41.0) that contains 77 mouse cell line profiles. Given values correspond to the number of repeats that are found at the analyzed STR marker sites. When the two alleles have the same number of repeats at a specific variant site, only one number is given, while two numbers indicate that the two alleles of this STR marker are different in regard to repeat length. In the case that three numbers are listed, it is most likely that the respective variant region is amplified in the respective cell line.

findings through increased scientific rigor and transparency by authenticating key biological materials, including cell lines.

We have not performed a karyotype analysis for this cell line yet, which would be an additional alternate that could be used for cell authentication. Such analysis has been done previously in our laboratory for two other hepatic stellate cell lines from mouse and rat [17, 18]. The observed numerical and structural chromosomal abnormalities in these cell lines were proposed as an alternate strategy that could be used for cell authentication. In addition, there are many other methods for cell authentication available including isoenzyme analysis, next generation-based single nucleotide profiling, and DNA fingerprinting, but STR profiling has become accepted as the simplest method to identify cross-contamination and cell misidentification [19].

## Author Contributions

**Conceptualization:** Ralf Weiskirchen.

**Data curation:** Ralf Weiskirchen.

**Formal analysis:** David A. Brenner, Ralf Weiskirchen.

**Funding acquisition:** Ralf Weiskirchen.

**Investigation:** Ralf Weiskirchen.

**Methodology:** Ralf Weiskirchen.

**Project administration:** Ralf Weiskirchen.

**Resources:** Ralf Weiskirchen.

**Supervision:** Ralf Weiskirchen.

**Validation:** Steffen K. Meurer, Ralf Weiskirchen.

**Visualization:** Ralf Weiskirchen.

**Writing – original draft:** Ralf Weiskirchen.

**Writing – review & editing:** Steffen K. Meurer, David A. Brenner, Ralf Weiskirchen.

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
