## [Decision Letter · Decision Letter 0]

14 Jun 2022

PONE-D-22-11946Multiplex Short Tandem Repeat Profiling of Immortalized Hepatic Stellate Cell line Col-GFP HSCPLOS ONE

Dear Dr. Weiskirchen,

Thank you for submitting your manuscript to PLOS ONE. After careful consideration, we feel that it has merit but does not fully meet PLOS ONE’s publication criteria as it currently stands. Therefore, we invite you to submit a revised version of the manuscript that addresses the points raised during the review process.

We look forward to receiving your revised manuscript.

Kind regards,

Pankaj Bhardwaj, Ph.D.

Academic Editor

PLOS ONE

2.We note that the grant information you provided in the ‘Funding Information’ and ‘Financial Disclosure’ sections do not match. 

"The laboratory of R.W. is supported by the German Research Foundation (grants WE2554/13-1, WE2554/15-1, and WE2554/17-1)."

Reviewers' comments:

Reviewer's Responses to Questions

**Comments to the Author**

1. Is the manuscript technically sound, and do the data support the conclusions?

Reviewer #1: Yes

Reviewer #2: Yes

Reviewer #3: Yes

2. Has the statistical analysis been performed appropriately and rigorously? 

Reviewer #1: Yes

Reviewer #2: Yes

Reviewer #3: N/A

3. Have the authors made all data underlying the findings in their manuscript fully available?

Reviewer #1: Yes

Reviewer #2: Yes

Reviewer #3: Yes

4. Is the manuscript presented in an intelligible fashion and written in standard English?

Reviewer #1: Yes

Reviewer #2: Yes

Reviewer #3: Yes

5. Review Comments to the Author

Reviewer #1: The research article entitled ‘Multiplex Short Tandem Repeat Profiling of Immortalized Hepatic Stellate Cell line Col-GFP HSC’ by Steffen K. Meurer et al provides STR profile for identification of mouse cell line termed as-Hepatic Stellate Cell line Col-GFP HSC. This cell line is an important tool to investigate fibrogenic signalling especially using PDGF, TGF-β1 etc. The reported information in this paper is useful to wider researcher community working with the different mouse/human cell lines. Therefore, I recommend the paper for publication, however following issues need to be addressed.

1. It is difficult to read the figure1 therefore use better resolution of figure 1.

2. It may be advised to rephrase the sentence ‘In this study… cell line authentication (line 15-16)’. It gives impression that this sentence is refering to PLoS One paper in the year 2013.

3. Line 58 check spelling of ‘stimuli’.

4. Table 1- What does the ‘allele range mean’? It may be mentioned in the table legends or may be inserted in the text somewhere.

5. Table 1- if the information on the primers and PCR size is available/ or in the public domain then it may be inserted.

6. Insert some statement on the significance of STR profile homologies. Refer line 123-127.

7. Provide the information on what the different numbers in the table 2 refer to? and what they signify?

Reviewer #2: This is a nice efforts by authors to authenticate their own cell line. However, there are serious lapses in introduction and discussion part.

1. Authors may discuss pros and cons of various cell line characterization methods, perhaps a tabular comparison will be nice.

2. there is no karyotyping or any other supporting data regarding the cell line. How does the karyotypes look in low passage versus high passage cells, specially focusing on any major abnormalities. it would be very nice if authors could include 2 karyotypes.

3. There is no discussion about the results. please elaborate. Also readers would like to know if you have/planned to develop any PCR multiplex for the characterization of your cell line.

Reviewer #3: Major comments

The authors have profiled the cell line using STR markers which will help to identify the cell line in future. However, this information alone is not sufficient enough to be published as an independent article. This needs to be clubbed with some additional data such as comparison between cell lines of same origin or other related cell lines used by researchers and the identification of any unique marker(s) that will help to differentiate these cell line from one another.

Minor comments

Repetition of text at several places.

6. PLOS authors have the option to publish the peer review history of their article (what does this mean?). If published, this will include your full peer review and any attached files.

Reviewer #1: **Yes: **Dr. Ram Gopal Nitharwal

Reviewer #2: **Yes: **sandeep singh

Reviewer #3: No

---

## [Author Response · Author response to Decision Letter 0]

15 Jul 2022

Reviewer #1

The research article entitled ‘Multiplex Short Tandem Repeat Profiling of Immortalized Hepatic Stellate Cell line Col-GFP HSC’ by Steffen K. Meurer et al provides STR profile for identification of mouse cell line termed as-Hepatic Stellate Cell line Col-GFP HSC. This cell line is an important tool to investigate fibrogenic signalling especially using PDGF, TGF-β1 etc. The reported information in this paper is useful to wider researcher community working with the different mouse/human cell lines. Therefore, I recommend the paper for publication, however following issues need to be addressed.

Dear Dr. Nitharwal,

Many thanks for the thoughtful review of our work and your critique. We have addressed your comments as stated in the point-to-point response. In the revised version we have marked all changes in red letters.

1. It is difficult to read the figure1 therefore use better resolution of figure 1.

We fully agree with your concern. In the pdf file that you reviewed the overall quality of figure 1 was insufficient. As suggested, we have increased the resolution of respective figure and hope that it now looks better in the pdf file that is manually produced by the online submission system. However, please note that the resolution in the pdf file generated by the online system is set to low resolution. For final quality refer to the TIF file of the image.

2. It may be advised to rephrase the sentence ‘In this study… cell line authentication (line 15-16)’. It gives impression that this sentence is referring to PLoS One paper in the year 2013.

Thanks for this suggestion. We have rephrased the mentioned sentence to “In the present study …..”.

3. Line 58 check spelling of ‘stimuli’.

We are sorry for this spelling error and corrected the term in the revised version.

4. Table 1- What does the ‘allele range mean’? It may be mentioned in the table legends or may be inserted in the text somewhere. 

That is a great suggestion. We have now explained the term “allele range” in the footnote of table 1 (now table 2).

5. Table 1- if the information on the primers and PCR size is available/ or in the public domain then it may be inserted.

We have now compiled a new table in which we have added the primer information. In the new Table 1 we have added a footnote about the sources of primer sequences. All other tables were lined up.

6. Insert some statement on the significance of STR profile homologies. Refer line 123-127.

This is an important comment. For computation of the significance of a STR profile, we used the Tanabe score, which is also known as the Sørensen-Dice coefficient (see Capes-Davis et al., 2012). This score is used to gauge the similarity of two samples and is defined as follows:

Percent match = (number shared alleles x 2) / (total number of alleles in the questioned profile + total number of alleles in the reference profile)

The similarity can range between 0 (completely different) and 1 (identical).

The highest homology of Col-GFP HSC was found to cell line AT-3 (score= 0.8293).

To clarify this issue we have added a short sentence explaining the Tanabe score in the Material and method section and added the respective reference (new reference 10).

7. Provide the information on what the different numbers in the table 2 refer to? and what they signify?

The different numbers refer to the allele constellation at respective loci in the mentioned cell lines. We have added a short comment on that issue in the footnote of respective table (now table 3).

Reviewer #2

This is a nice efforts by authors to authenticate their own cell line. However, there are serious lapses in introduction and discussion part.

Dear Dr. Singh,

Many thanks for the thoughtful review of our work and your critique. We have addressed your comments as stated in the point-to-point response. In the revised version we have marked all changes in red letters.

1. Authors may discuss pros and cons of various cell line characterization methods, perhaps a tabular comparison will be nice.

We not attempt to provide a review about cell line characterization methods. However, to address your comment, we have added some brief comments at the end of the discussion section. In addition, we have added a reference in which pros and cons of various cell line characterization methods are discussed.

2. there is no karyotyping or any other supporting data regarding the cell line. How does the karyotypes look in low passage versus high passage cells, specially focusing on any major abnormalities. it would be very nice if authors could include 2 karyotypes.

We have recently published two other papers in which we provide karyotypes for the murine cell line GRX (Schröder et al., Cells 2022;11(9):1504) and the rat cell line HSC-T6 (Nanda et al., Cells 2022;11(11):1783). We have not made any attempt to provide a karyotype analysis in this paper because we have now deposited this cell line in a nonprofit cell repository (DSMZ, https://www.dsmz.de/). They will now further characterize this cell line before sending them out to laboratories who request this cell line. However, they requested from us to provide a STR profile that can be used as a reference for this cell lines. This was done in our paper without providing information about the karyotype. However, to address your concern we have added a short comment on the necessity of a karyotype analysis at the end of the discussion section.

3. There is no discussion about the results. please elaborate. Also readers would like to know if you have/planned to develop any PCR multiplex for the characterization of your cell line.

The assay that was used for genotyping is a PCR multiplex assay. We have added the respective primers used in this assay in new table 1. All other tables were lined up.

Reviewer #3

Major comments

The authors have profiled the cell line using STR markers which will help to identify the cell line in future. However, this information alone is not sufficient enough to be published as an independent article. This needs to be clubbed with some additional data such as comparison between cell lines of same origin or other related cell lines used by researchers and the identification of any unique marker(s) that will help to differentiate these cell line from one another.

Many thanks for the thoughtful review of our work and your critique. It is a pity that you are not as enthusiastic as the other two reviewers. We hope that we can convince you with the following presentations that our paper is worth to be published.

As you might know, the animal welfare law has dramatically changed in the EU. Therefore, there is an urgent need to have well-characterized cell lines for biomedical research. This is the reason why we recently established the genetic characteristics of murine hepatic stellate cell line GRX (Schröder et al., Cells 2022;11(9):1504) and rat hepatic stellate line HSC-T6 (Nanda et al., Cells 2022;11(11):1783). As we discuss in our short report, the cell line Col-GFP HSC was established in our lab several years ago. After all these years, we now receive many requests for this cell line. In particular, several scientists will use this cell line in cell tracking experiments, which is very simple with our cells because they carry a GFP reporter cassette.

The STR profile established in the present study is unique and allows rapid authentication of this cell line. We want to prevent that somebody use unauthenticated cells from our laboratory. As you might know the actual Register of Misidentified Cell Lines currently lists already 576 cell lines that are misidentified (https://iclac.org/databases/cross-contaminations/). It is well accepted that misidentified or contaminated cell lines will waste valuable time and resources on experiments and provide the basis for flawed publication. With the STR profile that we have established everybody working with this cell line will be now able to provide the required cell authentication that is requested by many journals and funding agencies. This is the intention of our short communication.

We hope you can follow us in our argumentation.

Minor comments: Repetition of text at several places.

We have tried to remove unnecessary redundancies.

---

## [Decision Letter · Decision Letter 1]

22 Aug 2022

PONE-D-22-11946R1Multiplex Short Tandem Repeat Profiling of Immortalized Hepatic Stellate Cell line Col-GFP HSCPLOS ONE

Dear Dr. Weiskirchen,

Thank you for submitting your manuscript to PLOS ONE. After careful consideration, we feel that it has merit but does not fully meet PLOS ONE’s publication criteria as it currently stands. Therefore, we invite you to submit a revised version of the manuscript that addresses the points raised during the review process.

We look forward to receiving your revised manuscript.

Kind regards,

Pankaj Bhardwaj, Ph.D.

Academic Editor

PLOS ONE

Journal Requirements:

Reviewers' comments:

Reviewer's Responses to Questions

**Comments to the Author**

1. If the authors have adequately addressed your comments raised in a previous round of review and you feel that this manuscript is now acceptable for publication, you may indicate that here to bypass the “Comments to the Author” section, enter your conflict of interest statement in the “Confidential to Editor” section, and submit your "Accept" recommendation.

Reviewer #1: All comments have been addressed

Reviewer #3: (No Response)

Reviewer #4: (No Response)

2. Is the manuscript technically sound, and do the data support the conclusions?

Reviewer #1: Yes

Reviewer #3: Yes

Reviewer #4: Partly

3. Has the statistical analysis been performed appropriately and rigorously? 

Reviewer #1: N/A

Reviewer #3: N/A

Reviewer #4: N/A

4. Have the authors made all data underlying the findings in their manuscript fully available?

Reviewer #1: Yes

Reviewer #3: Yes

Reviewer #4: Yes

5. Is the manuscript presented in an intelligible fashion and written in standard English?

Reviewer #1: Yes

Reviewer #3: (No Response)

Reviewer #4: Yes

6. Review Comments to the Author

Reviewer #1: The revised version of the manuscript 'Multiplex Short Tandem Repeat Profiling of Immortalised Hepatic Stellate Cell line Col-GFP HSC' addresses all the specific concerns/points that were raised. The manuscript is recommended for publication.

Reviewer #3: It was quite unprofessional to use such a language. There is no need to take pity on anyone. My comments were professional and I stand by my comments. I have not questioned the quality of this study but didn't find the study enough to be published as an independent article. If others have recommend your article for publication as such it is completely their choice.

Reviewer #4: Steffen Meurer and colleagues improved their manuscript entitled “Multiplex Short Tandem Repeat Profiling of Immortalized Hepatic Stellate Cell line Col-GFP HSC” (PONE-D-22-11946R1) and addressed major points raised by the reviewers. There are only a few minor aspects that the authors should address to further improve their manuscript:

General remark: An own analysis of the STR by the authors would have been better for this paper, because then a detailed protocol for the STR analysis would be available for the readers in the section "Material and Methods".

A) Line 64: “for the cell line” (was used twice in one sentence)

B) Line 78: A 100 mm² culture dish does not exist and is certainly too small for 8 million cells. Probably a 10 cm (diameter) is meant here(?).

C) Line 81: “of Col-GFP HSC cells” (without cells). The abbreviation already contains the word cells.

D) Line 89: The term “PIGtail” must be defined and explained. Is it a 7-nucleotide tag?

E) Line 113: The term SV40 was already mentioned in the section “Introduction” (line 58). The abbreviation in the line 113 now differs from the former. A detailed description of SV40T should be given when mentioned first.

F) Line 154: Analysis of “interspecies contamination” was mentioned several times in the manuscript but was not shown by the authors. At least one appropriate citation should be given by the authors at this point.

7. PLOS authors have the option to publish the peer review history of their article (what does this mean?). If published, this will include your full peer review and any attached files.

Reviewer #1: **Yes: **Ram Gopal Nitharwal

Reviewer #3: No

Reviewer #4: No

---

## [Author Response · Author response to Decision Letter 1]

23 Aug 2022

Reviewer #1:

The revised version of the manuscript 'Multiplex Short Tandem Repeat Profiling of Immortalised Hepatic Stellate Cell line Col-GFP HSC' addresses all the specific concerns/points that were raised. The manuscript is recommended for publication.

Thanks again for all your input.

Reviewer #3:

It was quite unprofessional to use such a language. There is no need to take pity on anyone. My comments were professional and I stand by my comments. I have not questioned the quality of this study but didn't find the study enough to be published as an independent article. If others have recommend your article for publication as such it is completely their choice.

Dear reviewer #3,

We fully understand your displeasure. However, we had not the intension of taking pity on anyone. Quite the opposite, we are really grateful that you and the other reviewers helped to improve the quality of our work. We respect your opinion that you think that our study does not provide significant data to allow its publication. We agree that we only show a STR profile for the cell line that we established several years ago. In this regard our study is a small “add-on” to the previous paper that was published already nearly a decade ago (Meurer et al., PLoS ONE 2013;8(2):e56116). I guess everybody will realize that our paper is not a classical full paper but just a kind of addition. Nevertheless, this addition is very important because we have now deposited this cell line in a nonprofit cell repository (DSMZ, https://www.dsmz.de/). This will allow every researcher to obtain this cell line for research purposes. However, it is now well-accepted that a defined STR profile that allows easy cell authentication is mandatory when working with a cell line. This is the reason why we would like to publish our addition in the journal in which we originally described the generation of this continuous growing cell line.

We hope you understand our argumentation and are sorry if you think that we wanted to tackle you.

Reviewer #4:

Steffen Meurer and colleagues improved their manuscript entitled “Multiplex Short Tandem Repeat Profiling of Immortalized Hepatic Stellate Cell line Col-GFP HSC” (PONE-D-22-11946R1) and addressed major points raised by the reviewers. There are only a few minor aspects that the authors should address to further improve their manuscript:

Dear reviewer #4,

Many thanks for the time you spent in reading our short article. We are grateful for your thoughtful comments. In the revised version, we have addressed your minor comments and marked all changes in red letters.

General remark: An own analysis of the STR by the authors would have been better for this paper, because then a detailed protocol for the STR analysis would be available for the readers in the section "Material and Methods".

Yes we initially thought about doing that. However, we then realized that the customized 19 STR markers enclosed in the CellCheck Mouse 19 panel that were used in our study are essentially those that were proposed by the Consortium for Mouse Cell Line Authentication to validate STR markers for intraspecies identification of mouse cell lines (Almeida et al., PLoS ONE 2019;14(6):e0218412). This panel is nowadays widely applied in many laboratories and provides information on all markers that are necessary to allow a quick comparison with STR profiles form other cell lines at the Cellosaurus STR database (https://web.expasy.org/cellosaurus-str-search/). In our view, offering a protocol of our own would be rather confusion that purposeful. 

A) Line 64: “for the cell line” (was used twice in one sentence)

We have rephrased the mentioned sentence in line 64.

B) Line 78: A 100 mm² culture dish does not exist and is certainly too small for 8 million cells. Probably a 10 cm (diameter) is meant here(?).

You are correct. It must read “10 cm dish”. We have corrected this error.

C) Line 81: “of Col-GFP HSC cells” (without cells). The abbreviation already contains the word cells.

Thanks for this comments. We have deleted “without cells”

D) Line 89: The term “PIGtail” must be defined and explained. Is it a 7-nucleotide tag?

Yes, in the protocol it refers to a 7-nucleotide tag with the sequence “GTTTCTT”. According to your advice, we have added this information.

E) Line 113: The term SV40 was already mentioned in the section “Introduction” (line 58). The abbreviation in the line 113 now differs from the former. A detailed description of SV40T should be given when mentioned first.

Many thanks for this comment. In the revised version, we have now introduced the abbreviation SV40T upon first mentioning and harmonized the abbreviation to SV40T for Simian virus large T antigen.

F) Line 154: Analysis of “interspecies contamination” was mentioned several times in the manuscript but was not shown by the authors. At least one appropriate citation should be given by the authors at this point.

Yes this is correct. We have added a new reference (Masters et al., Proc Natl Acad Sci U S A 2001;98:8012-7) and again cited the two references of Ameida et al. (Refs 6 and 7) that discuss that STR profiling is suitable to detect interspecies contamination in human and mouse cell lines.

---

## [Editor Report · Decision Letter 2]

24 Aug 2022

Multiplex Short Tandem Repeat Profiling of Immortalized Hepatic Stellate Cell line Col-GFP HSC

PONE-D-22-11946R2

Dear Dr. Weiskirchen,

We’re pleased to inform you that your manuscript has been judged scientifically suitable for publication and will be formally accepted for publication once it meets all outstanding technical requirements.

Kind regards,

Pankaj Bhardwaj, Ph.D.

Academic Editor

PLOS ONE
---

## [Editor Report · Acceptance letter]

26 Aug 2022

PONE-D-22-11946R2 

Multiplex Short Tandem Repeat Profiling of Immortalized Hepatic Stellate Cell line Col-GFP HSC 

Dear Dr. Weiskirchen:

I'm pleased to inform you that your manuscript has been deemed suitable for publication in PLOS ONE. Congratulations! Your manuscript is now with our production department. 

Kind regards, 

on behalf of

Dr. Pankaj Bhardwaj 

Academic Editor

PLOS ONE